# In-Field Rainwater Harvesting Tillage in Semi-Arid Ecosystems: II Maize–Bean Intercrop Water and Radiation Use Efficiency

**DOI:** 10.3390/plants12162919

**Published:** 2023-08-11

**Authors:** Weldemichael Tesfuhuney, Muthianzhele Ravuluma, Admire Rukudzo Dzvene, Zaid Bello, Fourie Andries, Sue Walker, Davide Cammarano

**Affiliations:** 1Department of Soil, Crop, and Climate Sciences, University of the Free State, Bloemfontein 9301, South Africa; 2Agricultural Research Council (ARC), Soil, Climate, Water (SWC), Pretoria 0083, South Africa; 3Risk and Vulnerability Science Centre, Faculty of Science and Agriculture, University of Fort Hare, Alica 5700, South Africa; 4Agricultural Research Council (ARC), Grains Crops, Potchefstroom 2520, South Africa; 5Department of Agriculture and Rural Development (DARD), Glen, Bloemfontein 9360, South Africa; 6Department of Agroecology, Aarhus University, 8000 Aarhus, Denmark; davide.cammarano@agro.au.dk

**Keywords:** evapotranspiration, radiation interception, radiation use efficiency, soil water balance, water productivity

## Abstract

The purpose of this study was to evaluate alternative management practices such as in-field rainwater harvesting (IRWH) and intercropping techniques through conducting on-farm demonstrations. Seven homestead gardens in Thaba Nchu rural communities in the central part of South Africa were selected as demonstration trials. Two tillage systems, conventional (CON) and IRWH, as the main plot, and three cropping systems as sub-plot (sole maize and beans and intercropping) were used to measure water use and radiation use parameters. The water productivity (WP) of various treatments was positively related to the radiation use efficiency (RUE), and the degree of associations varied for different tillage systems. The water use in IRWH was higher by 15.1%, 8.3%, and 10.1% over the CON for sole maize and beans and intercropping, respectively. Similarly, the intercropping system showed water use advantages over the solely growing crops by 5% and 8% for maize and by 16% and 12% for beans under IRWH and CON tillage, respectively. Maximum RUE was found for sole maize and beans under IRWH, higher by 13% and 55% compared to the CON tillage, respectively. The RUE under IRWH tillage was estimated to be 0.65 and 0.39 g DM MJ^−1^ in sole maize and intercropping, respectively. However, in sole and intercropped beans, the RUE showed higher values of 1.02 g DM MJ^−1^ and 0.73 g DM MJ^−1^, respectively. WP and RUE were associated with water deficits and proportional to lower radiation use. This relationship indicates that the intercepted radiation by plants for photosynthesis is directly related to the transpiration rate until radiation saturation occurs. Therefore, the higher water deficit and lesser efficiency in using the radiation available during the season can be improved by practicing IRWH techniques. Furthermore, in semi-arid areas, to enhance the efficiency of water and radiation usage in intercropping management, it is crucial to adjust plant population and sowing dates based on water availability and the onset of rainfall.

## 1. Introduction

Improving both water use efficiency (WUE) and radiation use efficiency (RUE) in mixing cropping systems is crucial for increasing crop yields in dryland agriculture [1,2]. The crops in an intercropping system use resources differently, complementing each other and collectively producing higher yields than when grown individually in the same area [3,4].

Cereal–legume intercropping has several benefits, including increased yield [5], improved soil properties [6], and increased nitrogen-fixing bacteria [7,8]. Organized intercropping systems can also make better use of resources such as light, heat, water, and nutrients, resulting in higher yields and more stable crop groups [9,10]. However, to avoid negative impacts on crop growth, factors such as legume species selection, seeding rate, sowing date, and row spacing must be considered to limit competition between legumes and primary crops [11]. Improving rainwater productivity is one of the outstanding strategies for use in dryland farming; however, much of the productive rainwater is lost through ex-field runoff (*R_off_*) and soil evaporation (*E_s_*), resulting in extremely low productivity [12]. Oweis et al. [13] suggested that in dryland agriculture, over 50% of lost water could be recovered through improved water-harvesting techniques. Farmers in the semi-arid areas have therefore developed strategies, including in-field rainwater harvesting (IRWH), to cope with these uncertain and erratic rainfall patterns.

In the semi-arid crop production areas in the central part of South Africa, the problem of low and erratic rainfall is exacerbated by two major factors, viz., high runoff and high atmospheric evaporative demand [14,15], which lead to high evaporation of water from the soil surface. These losses hamper the efficient use of available water for crop production, and water losses need to be minimized to optimize rainwater productivity. Hence, the approach of IRWH with appropriate cultural management practices such as intercropping [14,16,17,18,19] is an important consideration for rainfed agriculture and can be used as an adaptation strategy against changes and variability of climate [20].

Intercropping is more productive than monocropping because it makes efficient use of resources such as water, nutrients, labor, and land, while also reducing pests, diseases, and weed infestation [21]. In particular, cereal and legume intercropping are recognized as a common cropping system throughout tropical developing countries [22,23]. In cereal–legume intercropping, cereal crops form relatively higher canopy structures than legume crops, and the roots of cereal crops grow to a greater depth than those of legume crops [24]. This indicates that the component crops probably have different spatial and temporal use of environmental resources such as radiation, water, and nutrients [25]. Therefore, integrating the techniques of rainwater harvesting (such as IRWH) and cereal–legume intercropping on smallholders’ arable fields or homestead gardens may improve productivity through efficient use of resources (water, radiation, and nitrogen). There are many such studies conducted on water use and radiation use on experimental stations, but very scarce obtain comparable results from on-farm trials with farmers utilizing common cultural practices.

In crops grown solely, there is a relationship between RUE and WUE, as reported in studies by Caviglia and Sadras [26], Sadra and Roget [27], Sadras [28] and Zhou et al. [29]. Key physiological and agronomic aspects of intercropping have also been widely investigated [24,30,31,32]. There are limited reports, however, comparing capture and efficiency in the use of resources (water and radiation) and their relationship with maize–bean sole and intercropping under the IRWH technique as compared to conventional (CON) tillage. Furthermore, most research is conducted in agronomic trials. However, on-farm research can provide innovative solutions for local villages in developing countries where funding is limited. In this research, we used an on-farm plot demonstration approach to explore the relationship between WUE and RUE and the mechanism behind it during a typical dry season in semi-arid environments. This study, therefore, hypothesizes that maize–bean intercropping under the Improved tillage (IRWH) system increases resource productivity and efficiency compared to solely grown crops. Additionally, there are positive relationships between WP and RUE in both IRWH and CON tillage systems, with higher water deficit and lesser available radiation use in CON compared to IRWH.

## 2. Materials and Methods

### 2.1. Study Area and Experimental Design

The study area (Thaba Nchu) is situated at a latitude of 29°12′33.6″ S, longitude 26°50′20.3″ E, and altitude of 1516 m, about 65 km from Bloemfontein in the Free State Province of South Africa. The two selected target study areas (Paradys and Morago villages) are located on the northern side, approximately 8 km from Thaba Nchu. Growing vegetables and rearing livestock in backyards is a common practice of smallholder households. Paradys and Morago villages have 271 and 300 ha of arable land as well as 1795 and 1650 ha of communal grazing area, respectively. Each household has access to about 2 to 4 ha of arable land. Moreover, households have 0.25–0.50 ha of residential land, a portion of which can be used as homestead gardens on which a household can produce crops such as maize, legumes, and vegetables and, to some extent, forage for their livestock. The clay loam soils of the demonstration plots belong to Sapane ecotope. The basic soil morphological properties are deep dark brown and brown–grey-black, for Paradys and Morago, with A horizon of clay loam having a particle size of clay 34.0% and 29.4%, respectively. The demonstration trials were conducted on seven household homestead gardens. Accordingly, there were two tillage systems as the main treatment (IRWH and CON) and three cropping systems as sub-treatments (sole maize, sole beans, and maize–beans intercropping). Each combination treatment was replicated in four and three demonstration plots in Paradys and Morago villages, respectively. 

### 2.2. Field Measurements 

For this experiment (2018/19 growing season), all 6 treatments (2 tillages × 3 cropping systems) were used to measure soil water and radiation canopy interception and to calculate water productivity (WP). Furthermore, crop growth parameters, the grain, and biological yield (above-ground dry matter, AGDM) values were used to compare the two tillage systems for sole maize and beans and intercropping systems. Detailed description of the site, land preparation, runoff and basin area construction, farmers’ cultivar choice, cropping season, and crop management aspects are described in WRC 2020, Report No. K/2821/4 [33]. 

Destructive sampling was used to determine the leaf area and biomass of the maize and bean crops. For the leaf area, the fully matured leaves were cut off the crops, and the leaf area measurement was determined using a LI3000 leaf area meter (LI-COR Inc., Lincoln, NE, USA). The AGDM and leaf area were measured periodically from 20 days after emergence (DAE) until the plants attained maximum size (85 DAE). During sampling, the height of each plant was recorded, cut at the soil surface, and then separated into green and dead leaves, stems, and reproductive organs. To determine the final AGDM, a sample quadrant of 4 m^2^ with three replications from each treatment was harvested at the end of the season. To determine the harvested biomass, samples were dried in an oven regulated at 70 °C for 72 h. Thus, the AGDM, partitioned into leaf, stem, and reproductive organs, was calculated as oven-dry material in kg ha^−1^.

### 2.3. Water Balance Components

A simple form of water balance quantification appropriate for IRWH and CON in arid and semi-arid areas was adopted from Hillel [34]. Evapotranspiration (*ET*) can be estimated with the water balance equation for dryland crop production in soils without a water table and without significant internal lateral water movement and can be written as follows [35]: Water for yield=water gains−water losses
(1)$$ET=\left(P\pm \Delta S\right)-(R_{off}+D)$$

The equation states the general concept that water for yield is equal to the water gains minus water losses. In this model, a portion of rainfall (*P*) infiltrates into the soil and becomes available for root extraction, together with the change in soil water content (Δ*S*) between the beginning and end periods of the growing season. The losses include the amount of water evaporated from the soil surface and plant transpiration (*ET*), the surface runoff (*R_off_*), and the drainage amount (*D*). The IRWH technique has two different sections in each field, the basin and runoff area, which are practically linked as the runoff strip feeds water into the basins as run-on (*R_on_*), while the CON tillage is exposed to ex-field runoff losses. 

Measurement of soil water content (*θ_r_*) was performed in the demonstration plots. To monitor *θ_r_*, neutron water meter steel access tubes were inserted to a depth of 1500 mm—that is, to a depth greater than the expected roots. Soil water content was measured at an interval of 1–2 weeks to a depth of 1500 m using a neutron water meter (NWM, Campbell Pacific Nuclear model 503, Antioch, CA, USA, 1994) to take neutron counts down the access tubes. Measurements of *θ_r_* were carried out during the growing season at 300 mm depth intervals starting at 150 mm (being 150, 450, 750, 1050, and 1350). This procedure ensures that the different pedological layers in the soil have been adequately represented. Due to the clay loam soil texture on the top surface and increasing clay content with depth down the profile on both sites, the soil is expected to reach a maximum water-holding value within the 600–900 mm layer. The high clay content below 900 mm reduces deep percolation, so drainage losses are considered to be negligible throughout this study. Due to that fact, the runoff data downloaded from the automated runoff tipper were incomplete and unreliable in terms of their inclusion in the water balance estimations. Thus, an empirical model developed by Anderson [36] for clay soil Glen Bonheim ecotopes was used to estimate the R_off_ amount during the growing season and equated as follows (Equation (2)):(2)$$R_{off}=0.2678P-2.5298$$
where *R_off_* = runoff, part of the amount of rainwater that could be a loss as ex-field runoff from the CON plots; *P* = amount of rain.

Water productivity (*WP*) was determined with an approach used by Passioura [37] **as** productivity is a function of the amount of rainwater during the growing season. *WP_g_,* therefore, measures the efficiency with which a particular crop can convert the water used by the plant into grain yield or biomass during a particular growing season: (3)$$WP={AGDM}_{g}/P_{g}\;(\mathrm{kg}\;{\mathrm{ha}}^{-1}\;{\mathrm{mm}}^{-1})$$
where *WP* is water productivity, and *P_g_* is the amount of rain during the growing season in mm.

Water use efficiency (*WUE*) was used to measure the efficiency with which a particular crop can convert the water available during the growing season [17,38,39,40]. Thus, *WUE_ET_* was determined with a slightly modified version of Hillel [38], Passioura [41], and Tanner and Sinclair [39] as follows:(4)$$WUE=Y_{g}/ET\;(\mathrm{kg}\;{\mathrm{ha}}^{-1}\;{\mathrm{mm}}^{-1})$$
where *WUE* is the water use efficiency in terms of total evapotranspiration (*ET*) in mm.

### 2.4. Radiation Canopy Interception 

The sub-treatment included two cropping systems that affected the microclimate, namely, sole cropping with no shading and intercropping with a shading effect and resource use competition. Moreover, the different tillage systems also differ in canopy configuration, which may affect the radiation interception of the crop canopies. The photosynthetic active radiation (PAR 0.4–0.7 μmol) was measured above and beneath the plant canopy with a single 1 m line quantum sensor (Kipp & Zonen model PQS 1, LI-COR models LI-190) that was set perpendicular in between the cropping rows. The line quantum sensor was placed in between the maize, beans, and maize–beans intercrop rows at the soil surface and above the canopy. The PAR measurement was taken at an interval of 7–15 days throughout the growing season. The PAR was measured around midday between 12:00–14:00 h South African Standard Time (SAST). 

The fraction of radiation intercepted by crop canopy (*F*) was estimated on the bases of Beer’s laws [42]:

(i)In maize/beans intercropping, the lower canopy layer consists of both maize and beans layers, while the upper layer only includes maize. Incident solar radiation at the top of the intercropping bean canopy is equivalent to F by the maize in the upper (*F_MU_*). This will be estimated by using a simple equation (Adopted from Tsubo and Walker, 2002) (Equation (5)):

(5)$$F_{MU}=1-\exp\left(-K_{m}L_{MU}\right)$$
where *K_m_* is the canopy extinction coefficient for maize and *L_MU_* is the LAI with uniform leaf density in the upper canopy (Equation (6)):(6)$$L_{MU}=\frac{h_{M}\;h_{B}}{h_{m}}T_{LM}$$
where *T_LM_* is the total maize leaf area, and *h_M_* and *h_B_* are the height of maize and beans canopy.

(ii)To measure radiation intercepted by each component of the crops in intercropping, a partitioning equation adopted from Tsubo and Walker [43] was used. Therefore, the fraction of radiation intercepted by beans (*F_B_*) and the fraction of maize at the lower layer (*F_ML_*) was estimated as follows (Equations (7) and (8)):

(7)$$F_{B}=\frac{K_{B}L_{B}}{K_{B}L_{B}+K_{M}L_{ML}}F_{M/B}$$(8)$$F_{ML}=\frac{K_{M}L_{ML}}{K_{B}L_{B}+K_{M}L_{ML}}F_{M/B}$$
where:-*K_B_* and *K_M_* is the canopy extinction coefficient for beans and maize (according to Tsubo and Walker [43], it was estimated at 0.64 and 0.43, respectively).-*L_ML_* and *L_B_* are maize and bean LAI in the lower canopy layer.-*F_M/B_* is a fraction of radiation interception by the crops of maize and beans in the lower canopy layer. This is equivalent to the difference between overall F by the intercrop and *F* by maize in the upper layer.

(iii)The LAI in the lower maize layer in the intercropping and the total intercepted was radiation estimated as Equation (9):



(9)
$$L_{ML}=\frac{h_{B}}{h_{M}}T_{LM}$$



Therefore, the fraction intercepted by maize crop includes both upper and lower (Equation (10)):(10)$$F_{M}=F_{MU}+F_{ML}$$

(iv)Radiation use efficiency (*RUE*) for beans and maize can be calculated as (Monteith, [44] (Equations (11) and (12)):

(11)$${RUE}_{B}=\frac{W_{B}}{I_{o}F_{B}}\;and\;{RUE}_{M}=\frac{W_{M}}{I_{o}F_{M}}$$(12)$${RUE}_{BM}=\frac{W_{BM}}{I_{o}F_{M}}$$
where *W_B_* and *W_M_* are dry matter (in kg) for beans and maize, respectively, and *I_o_* is the incident radiation in (MJ m^−2^ d^−1^). Comparative change in *RUE* was calculated according to Morris and Garrity [45] to relate productivity across varying cropping systems. The indice was based on relative rather than absolute values. Change in *RUE* was calculated based on dry matter, as shown in Equation (13).
(13)$$\Delta X=\left[\left(\frac{X_{Ic}}{P_{m}X_{sm}+P_{b}X_{sb}}\right)-1\right]\times 100$$
where X is *RUE*, P_m_ is the proportion of maize in intercrop, P_b_ is the proportion of beans in intercrop, subscripts Ic is intercrop, sm is sole maize, and sb is sole beans. Proportions of maize and beans in the intercrop are given by P_m_ = *D_m_/(D_m_ + D_b_*), with *D_m_* and *D_b_* being the density in intercropping relative to sole cropping of maize and beans, respectively. For interpretation, when Δ is greater than zero, it is assumed to be higher in the intercrop system relative to the sole crop.

### 2.5. Statistical Analysis 

Analysis of variance (ANOVA) was performed for the comparison of different treatments using SAS 9.1.3 for Windows (SAS Inst Inc., Singapore) [46]. When the significance of the treatment on the F-statistic is mentioned, it refers to a comparison using the least significant differences (LSD) at the 0.05 probability level. In this study, a relationship between RUE and WP was analyzed as the slope of the linear regression using aggregated data from the two locations for different cropping systems under different tillages to understand the effect of available soil water for productivity and the atmospheric demand in the semi-arid crop production system. 

## 3. Results

### 3.1. Leaf Area Index

In this study, the sole-cropped maize under CON had the highest LAI of 4.38 (Figure 1a). The lowest LAI that was obtained, 3.33, in maize intercropping under IRWH, which is contradictory to the expected outcome. Contrary to observations from Morago, in Paradys, the maize in the IRWH treatments exhibited higher LAI compared to CON treatments (Figure 1b). Throughout the study, the LAI of the intercropping maize values were not significantly different. The sole and intercropping treatments under CON tillage were lower than those in IRWH. Both sole and intercropped beans in IRWH treatment plots were significantly different throughout the study period. Except for intercropped beans in CON plots, 85 days after emergence, the LAI of beans from the rest of the treatments were not significantly different. The lowest LAI at the end of the study period was 1.49, found in intercropped beans in CON plots. Irrespective of the cropping systems, the plants from the IRWH tillage system plots had higher LAI as compared to the plants from the CON treatment plots. From 28 till 70 days after emergence, there was no significant difference between the cropping systems treatments under IRWH tillage systems. Under the CON tillage system, the sole- and intercropping treatments had LAI that showed no significant difference from 28 to 50 days after emergence.

### 3.2. Dry Matter Accumulation

In the village of Morago, during the early growing stage (38 DAE), there was significantly lower above-ground dry matter (DM) accumulation in sole cropping among the tillage systems compared to IRWH (Table 1). Only sole beans had a significantly higher (7.5 and 7.2 g m^−2^) DM accumulation compared to maize. In later growth stages, before the crops reached flowering, the maize crops showed no significant difference among the treatments, but higher DM values were observed for IRWH sole maize (31.8 g m^−2^ at 50 DAE). During tasselling, the DM accumulation of maize significantly increased in all treatments except for CON sole maize; however, at 85 DAE, when the crops started grain filling, the DM showed significant differences among all the treatments. In the village of Paradys, the DM was higher compared to Morago’s demonstration plots. In Paradys, the sole cropping under IRWH tillage showed significantly higher DM accumulation throughout the growing season among the treatments (except at 50 DAE). From flower initiation to grain filling (63–85 DAE), there were no significant differences in DM accumulation between both sole cropping (under IRWH and CON) systems and CON intercropped maize, but inconsistent measurements were noticed due to variation in growth after the long dry spells. However, in both villages, the CON intercrop maize showed higher DM accumulation during the growing season compared to IRWH. 

Similarly, with beans under IRWH, the Paradys demonstration plots showed higher DM accumulation compared to Morago village demonstration plots, while the CON beans showed higher DM in Morago compared to Paradys (Table 1). During the later stage, the DM accumulation revealed high variations in all treatments. Even though the IRWH sole beans showed significantly higher DM accumulation in Morago, there were no significant differences in DM among the treatments during grain filling, while the IRWH showed significantly greater DM accumulation compared to CON. In Paradys, significantly higher DM (812.3 g m^−2^) of maize and beans intercrop was found under IRWH tillage compared to CON, but in Morago, there was no significant difference of DM observed between the two tillage systems. 

### 3.3. Soil Water Balance Components

#### 3.3.1. Soil Water Content

The soil water content of IRWH was higher than that of CON in sole maize plots throughout the season (Figure 2). The differences in soil water content between the two tillages ranged from 18.2 mm to 58.3 mm for the whole season. However, in the sole beans and intercropping plots, IRWH was not higher than CON throughout the season. In the sole beans plot, 57 days after planting (29 March), both tillage systems recorded the same amount of soil water content. Meanwhile, by the end of the season, IRWH had recorded higher soil water content (365 mm) than CON (346 mm). The SWC was the same for both IRWH and CON for intercropping plots 52 days after planting (24 March). Soil water content was also higher in IRWH than in CON by the end of the season. These patterns suggest more investigation into soil water extraction patterns of crops in intercropping treatment plots. Nevertheless, IRWH had higher soil water content than CON at the beginning of the season in all the treatments. This showed a means of good water storage before the growing season. 

#### 3.3.2. In-Field Runoff

Runoff is classified into two different water-collecting aspects; these are: (i) water running out of the planted field and water harvested on the basin area under IRWH structures, viz., ex-field runoff (R_off_) and Run on (R_on_), respectively. The result in Figure 3 indicates the cumulative amount of ex-field runoff during the growing season, which is estimated to be a total amount of runoff of 83.6 mm. After the planting date, substantial rain was received at the end of January, which created 8 mm of runoff, 9.5% of the total runoff harvested during the growing season. However, there was a long dry spell period after that rainstorm. At the later stage of the crops, i.e., 55 DOY, 75 DOY, and after 86 DOY, the runoff was estimated at 27 mm (32%), 23 mm (27.5%), and 25 mm (29.0%), respectively. The rainfall distribution was very poor, and the IRWH plot had the advantage of using the stored water during those dry spell events. These results accentuated that, firstly, under the technique of IRWH from a 2 m runoff strip, it is possible to harvest about one quarter (27%) of the amount of rainfall, which could be a loss as ex-field runoff from the cropped field. This amount of water contributes to the productivity of water-scarce dryland farming in semi-arid areas. 

In the graph, it can be seen that the increase in R_on_ water in crucial growth stages, when the crops reached flowering and grain filling, influences the yield and productivity of the crops. The continuous small rain events also contribute to keeping the soil surface structure of the runoff strip stable and compacted to yield enough runoff. This shows the importance of small rain events in producing in-field runoff under varying surface treatments [15]. Many long-term statistical models [14,47,48,49] excluded small rain events to obtain a realistic R_on_ amount. However, Anderson et al. [50] concluded that statistical models provide a better estimation of runoff at low rainfall amounts, as their R^2^ using all data points was generally considered better than those with only rain amounts greater than 8 mm. Thus, it is considered that for long-term prediction, the inclusion of all sorts of rain events (small rains–higher events) is a valuable asset for the IRWH system. However, in estimating the in-field runoff for IRWH tillage, on top of the rainfall amount, it Is important to consider the rainfall intensity and duration, as well as surface treatments. 

#### 3.3.3. Estimation of Evapotranspiration 

Crop water use, also known as evapotranspiration (ET), is the water used by a crop for growth processes. ET is influenced by prevailing weather conditions, available water in the soil, crop species, and growth stage. The growth stages of these two partner crops (maize and beans) can be dived into different developmental stages. The growth stages of the maize and beans described in this study are almost identical to other previous studies (for example, that reported by FAO: Doorenbos and Kassam; FAO, 2000) [51,52]. For both crops, the growth stages can be divided into four phases: For maize—S-1 = initial vegetative phase, GS-2 = active vegetative phase, GS-3 = initial grain-filling phase, and GS-4 = active grain-filling phase.For beans—GS-1 = emergence and early vegetative growth, GS-2 = branching and rapid vegetative growth, GS-3 = flowering and pod formation, and GS-4 = pod fill and maturation.

According to the measurement dates, the four growth stages of the crops correspond to 0–40, 41–65, 66–85, and 86–125 days after planting, respectively. 

The water balance processes identified in Equation (1) are relevant in the functioning, productivity, and in explaining the soil–plant–atmosphere continuum (SPAC) under the CON and IRWH techniques. Thus, it is important to monitor these processes through field measurements and estimations of water balance components in order to obtain a good understanding of improved crop productivity for different cropping systems under different tillages. Evapotranspiration (ET) was estimated as residual using the water balance equation as determined by Equation (1). Table 2 shows the summary of the whole water balance components from the aggregated data of the two locations for the growing season (January–May 2019).

In the sole maize and beans and intercropping under IRWH, the ET was estimated at 340.4 mm, 301.8 mm, and 359.8 mm, respectively, during the growing season. Overall, the intercropping under CON tillage used the absolute highest ET. However, the IRWH with intercropping gave lower ET values than their respected solely grown beans and maize treatments. Moreover, ET from the CON tillage (262.4 mm) was obtained at a relatively lower with intercropping, but the sole beans under CON tillage produced lower ET compared to sole maize. The reason for this could lie in the effectiveness of additional water stored in the basin area and the ability of water to penetrate and infiltrate as well. Thus, all intercropped beans benefitted from the canopy shade on both sides, and the mean ET across the CON tillage was significantly larger than the other tillage system (IRWH). The CON tillage had an advantage in reducing soil evaporation through the shading effect compared to IRWH because the runoff strips between the tramlines are exposed to soil evaporation, while the basin area of the IRWH promotes infiltration to the crops’ root zone. 

### 3.4. Water Use

#### 3.4.1. Water Productivity (WP)

The overall results indicate that the WP varied between 15.12 and 8.34 and 10.10 and 5.34 kg ha^−1^ mm^−1^ for maize and beans AGDM, respectively (Table 3 (a)). The statistical results show that there were significant differences due to the effect of rainwater harvesting in basin areas on WP. There was a significant variation between the cropping systems in both crops. A different WP trend for Yg was observed viz. the maize sole (IRWH-S-M), significantly higher than both sole and intercropped maize (CON-S-M & CON-Ic-M), and the opposite water productivity was shown in beans with highest WP values in sole beans under IRWH (IRWH-S-B) compared to intercropped beans with no significant differences. Nevertheless, there was no significant variation observed between the treatments of beans.

#### 3.4.2. Water Use Efficiency (WUE)

The consideration of evapotranspiration in evaluating rainwater efficiencies may be able to show an advantage in practicing IRWH techniques for semi-arid climatic conditions. WUE was calculated using the residual ET from the water balance calculations from planting (January 2019) until harvest (May 2019). However, computing a reliable water use based on ET requires sufficient soil water content measurement that shows the changes based on the continuous plant roots’ water consumption and losses due to environmental factors. 

The results indicate that the WUE for AGDM varied between 13.06 and 9.87 and 10.40 and 6.44 kg ha^−1^ mm^−1^ during the growing season for different tillage and cropping system treatments for maize and beans, respectively (Table 3 (b)). The highest WUE of maize (AGDM) was found in the intercropped maize under the IRWH system, with no significant differences in both tillage systems (IRWH and CON) of the solely growing maize. Statistical analysis revealed that the IRWH tillage system on beans (AGDM) had a significant effect on the efficiency of water use as a function of evapotranspiration, with higher values in IRWH-S-M (10.40), and no significant differences were observed in both cropping systems under CON and in the intercropped beans under the IRWH. On the contrary, the function of evapotranspiration for WUE showed the significantly lowest values in the CON-intercrop (9.87 and 6.44 kg ha^−1^ mm^−1^ for maize and beans, respectively). 

With regard to Yg as a function of WUE, the results showed irregular trends, with higher values in sole-cropping compared to intercropping for both under IRWH and CON tillages. Nevertheless, neither the tillage nor the cropping systems show significant differences for both crops. However, one can consider maximizing crop WUE by using improved management; it is necessary both to conserve water and to promote maximal growth or minimize losses through runoff, evaporation, and transpiration. It also includes optimizing growing conditions by proper timing and performance of planting and harvesting, tillage systems, fertilization, and pest control. In short, raising water-use efficiency requires good farming practices from start to finish by collecting additional water to the root zone. Generally, but not always, the yield of cropping systems is proportional to total growth and to transpiration rather than applying an un-partitioned ET component in computing the water balance components in considering crop water use. Some inconsistencies were found in the degree of yield and total above-ground efficiency in terms of ET. From the results in Table 3, therefore, it can be seen that the WUE variations could be due to the dual effect of additional soil water and canopy shading. Moreover, the fluctuation in water use, therefore, must be a reflection of SWC measurement. However, these variations in efficiency should be examined further by applying water productivity as a function of transpiration.

### 3.5. Radiation Canopy Interception

The above-ground production of biomass can be closely related to light use efficiency but only marginally to intercepted radiation during the season and is mainly caused by management practices and canopy structures. The crop intercepted photosynthetically active radiation (PAR) and radiation use efficiency (RUE) vary markedly in cropping systems and/or tillage systems and are related to LAI and plant height. The RUE is another important factor for dry matter accumulation in addition to intercepted PAR. Differing from the previous studies in RWH techniques, in this study, some relationships were established among different cropping systems (sole and intercropping) with different canopy configurations in both CON and IRWH.

#### 3.5.1. Fraction of Intercepted PAR (fIPAR)

In the Morago site in both crops (Figure 4a,b), there was a high variation in the fraction of intercepted photosynthetic active radiation (fIPAR). In demonstration plots, the fIPAR for intercropped maize under CON (CON-Ic-M) peaked at 63 DAE (0.88), while the intercropped maize under IRWH (IRWH-Ic-M) peaked at 85 DAE (0.64). However, sole maize under CON and IRWH (CON-S-M and IRWH-S-M) peaked at 70 DAE, and the fIPAR was 0.60 and 0.52, respectively (Figure 4a). The fIPAR of CON-S-M and IRWH-Ic-M showed no significant differences throughout the measurement period during the growing season. Nevertheless, CON-Ic-M and IRWH-S-M showed the highest and the lowest fIPAR across the growing season (Figure 4a). This indicates the difference in canopy configuration and plant raw arrangement between the CON and IRWH tillage systems that influence the radiation interception by maize crops. 

In sole and intercropped beans, a different radiation interception amount was observed compared to maize treatments (Figure 4b). In all treatments, the fIPAR was higher in beans compared to maize under both tillage systems, with a higher value in CON intercropped beans (CON-Ic-B) and peak (0.93) at 85 DAE. Similarly, in IRWH sole and intercropped beans and CON sole beans (IRWH-Ic-B, IRWH-S-B, and CON-S-B, respectively), the fIPAR reached a peak between 70 and 85 DAE with fIPAR values of 0.70, 0.66, and 0.62. Similar to maize, the CON-S-B showed the highest fIPAR values compared to the other three treatments (IRWH-Ic-M, IRWH-S-B, and CON-S-B), but the CON-Ic-B showed higher values at the early stage until 50 DAE.

In the Paradys demonstration plots (Figure 5a,b), the fIPAR for sole maize under IRWH (IRWH-S-M) increased sharply from DAE 38 to 63 DAE, with the highest peak value of 0.77, but the sole maize under CON (CON-S-M) showed lower intercepted fraction and peaked at 50–63 DAE compared to IRWH (Figure 4a). However, both intercropped maize under CON and IRWH (CON-Ic-M and IRWH-Ic-M) peaked at 63 and 70 DAE, with fIPAR values of 0.49 and 0.64. There were large variations in fIPAR between sole and intercropped maize for both CON and IRWH tillages. In Paradys demonstration plots (Figure 5b), the intercepted fraction (fIPAR) of sole beans under IRWH (IRWH-S-B) increased slowly to the maximum interception (70% at 85 DAE), and the value of fIPAR was greater by 14%, 20%, and 9% compared to CON-Ic-B, CON-S-B, and IRWH-Ic-B, respectively. In general, the architecture of the canopy, which was affected by crop densities, crop height, and row arrangement, could be the deciding factor for crop intercepted PAR. The separation of the maize upper canopy in the intercropping led to more intercepted PAR of beans compared to the sole cropping. Moreover, the distance between maize tramlines under IRWH was also another issue for radiation interception advantageous to the increase in intercepted PAR for the short crop canopy. 

The graphical relationship between LAI and fIPAR is presented in Figure 6a–c. The LAI and fIPAR showed a logarithmic relationship with R^2^ values of 0.68, 0.54, and 0.69 for CON tillage and 0.51, 0.94, and 0.73 for IRWH in sole maize, sole beans, and intercropping, respectively. In all cropping systems, fIPAR increased with an increase in LAI, initially at a higher rate and then at a lower rate and finally flattening. Overall, with regard to trends, the sole cropping in both crops showed significant differences between the IRWH and CON tillages, with higher values in fIPAR in CON compared to the IRWH, with wide runoff strips between the tramlines. Notwithstanding the different canopy architecture in the intercropping with maize and beans, there were no significant differences between IRWH and CON in the relationship of LAI and fIPAR. Moreover, there was a high variation in fIPAR in CON plots compared to IRWH. In general, due to the selected date of measurement, the logarithmic relationships can be applied to eliminate some outliers due to changing light interception and/or unforeseen atmospheric conditions. 

#### 3.5.2. Total Intercepted Radiation (TPAR)

As part of the study, measurements of growth parameters were performed, such as leaf area index, above-ground biomass accumulation (Figure 1 and Table 1), and partitioning intercepted photosynthetic active radiation (IPAR) and calculated RUE as a function of DM accumulation (Table 4). In all treatments, the TPAR increased linearly with days after planting during the growing season in both experimental sites. The results in Table 4 (a) show the highest TPAR in Morago demonstration plots found for CON tillage with intercropped maize and beans treatments, with total values of 611.8 MJ and 800.4 MJ, respectively. The lowest TPAR was measured in IRWH tillage, with 385.9 MJ and 522.5 MJ for sole maize and some beans, respectively. At Paradys, the measurements from the IRWH showed a higher TPAR value, intercropped 582.0 MJ for intercropped maize and 655.9 MJ for sole beans, and the lowest TPAR was found in the CON tillage for maize sole treatment (Table 4 (b)). 

The results of RUE for all treatments and sites are also shown in Table 4. In comparing the results during the growing season, the RUE values varied and observed inconsistencies due to differences in growth stages and climatic conditions, like the cloud cover and timing of the measurements. For example, in the IRWH in sole maize and intercropping, the RUE ranges from 0.04 to 0.65 and 0.0 to 0.39 g DM MJ^−1^, respectively. However, in beans planted solely and intercropped, the RUE range was much wider, with values 0.11–1.31 g DM MJ^−1^ and 0.19–0.89 g DM MJ^−1^, respectively. In combining the RUE of maize and beans intercropping treatments, the highest RUE value was found at 85 DAE (1.34 g DM MJ^−1^) for IRWH (0.95 g DM MJ^−1^) and at 96 DAE for CON tillage. In general, the greater RUE was found in intercropping compared to sole cropping. A similar trend was observed in Paradys with higher RUE in intercropping, 1.35 and 1.12 g DM MJ^−1^ for IRWH and CON tillages. The results indicate the contribution of maize–bean intercropping under IRWH tillage, showing improvements in maize canopy size, radiation interception, and RUE. Thus, increased water availability through IRWH enhances the productivity of maize–bean intercropping and is closely associated with radiation use efficiency. 

The total TPAR of both maize and beans under IRWH showed the trend of sole > intercropping, but under CON tillage, the intercropped beans demonstrated higher DM production compared to IRWH. The highest AGDM under IRWH was observed in sole beans (738.6 g m^−2^) at the Paradys site. The CON sole beans in the Morago site demonstrated lower AGDM compared to intercropped. The dry matter of intercropping systems was higher than the solely grown crops. These results indicate the contribution of maize–bean intercropping under IRWH tillage, showing improvements in maize canopy size, radiation interception, and RUE. Thus, increased water availability through IRWH enhances the productivity of maize–bean intercropping and is closely associated with radiation use efficiency.

### 3.6. Relationship between Water and Radiation Use 

In this study, RUE (AGDM/TPAR) was plotted against WP (AGDM/P_g_) for sole- and intercropping systems under two different tillages (CON and IRWH), as shown in Figure 6a,b. In all cropping systems under both tillages, RUE increased as WP increased until radiation canopy interception reached the saturation level to produce biomass and then tended to be constant. Therefore, in agreement with the second hypothesis, the water use of various treatments was related to radiation use in rainfed semi-arid areas of the study site for maize–bean cropping systems under different tillage systems. However, it is expected to vary the degree of associations for different tillage techniques (i.e., between CON and IRWH). By pooling all maize and beans data together, a comparison was made between CON and IRWH for solely grown crops (Figure 6a) and maize–bean intercropped treatments (Figure 6b). The coefficient of determination (R^2^) of sole cropping shows higher values compared to intercropping, which is 0.84 and 0.88 vs. 0.67 and 0.82 for IRWH and CON, respectively. There were also significant differences (*p*
≤ 0.05) between the two tillages in both cropping systems. Furthermore, in the analysis, attempts were made to show the radiation saturation level with an increase in seasonal rainwater use for biomass production by the crops. The maximum RUE was calculated as an average of the three highest RUE of each data set (IRWH-Sole-M, IRWH-Sole-B, CON-Sole-M, and C-Sole-B) and (IRWH-Ic-M, IRWH-Ic-B, CON-Ic-M, and C-Ic-B), while RUE between zero and the maximum values was determined as the slope of linear regression with the zero intercepts (Figure 7a,b), and the results are summarized in Table 5.

In sole cropping, the maximum RUE was found under IRWH for solely grown maize and beans, which is higher by 13% and 55% than the CON tillage, respectively. In contrast, for the intercropping system, the maximum RUE was found to be lower for IRWH for only intercropped maize (by 19%), but the intercropped beans showed higher maximum RUE (by 12%) compared to CON tillage. This relationship indicates that the observed radiation by plants for photosynthesis is directly related to the transpiration rate until saturation occurs. Tsubo et al. (2003) described the high water requirement as a water deficit and proportional to lower RUE. Similarly, in this study, in sole cropping for maize and beans and intercropped beans, the CON showed a higher water deficit and lesser efficiency in using the radiation available during the season compared to IRWH. However, despite the advantage of IRWH over the CON, the intercropped maize showed a greater water deficit compared to CON tillage; this could be due to the higher competition of resources from the partner crop (beans) with shallow rooting structure for shallow soils in the study area. In general, in the IRWH, the maximum RUE was higher in the sole-cropping system, while in the CON tillage, a higher maximum RUE was observed in the intercropping system. 

## 4. Discussion

The IRWH system has been shown to improve soil water storage and productivity in dryland ecosystems. However, the ecophysiological responses of maize–bean intercropping under the IRWH system have not been studied comparatively. The high maize DM accumulation observed in the sole maize crop could be due to a lack of competition for resources such as light, nutrients, and water. Differences in root depth, lateral spread, and density can affect competition for nutrients in an intercropping [6,53]. Maize is usually taller with a faster-growing or more extensive root system, particularly a larger mass of fine roots, and is competitive for water and soil nitrogen [54]. The maize plants in the intercrops could have shadowed beans, reducing the amount of light required for growth. It is important to consider the number of maize copes and pods for beans when comparing sole- and intercropping under various tillage systems, as plant spacing varied accordingly. The increase in LAI is also attributed to foliage expansion because of the development of new leaves and the enlargement of existing leaves [55]. This could be ascribed to the lower rate of change of fIPAR to the higher rate of change of LAI after achieving the peaks of fIPAR and LAI, respectively [56].

The ability of crops to capture resources such as water, light, nutrients, and CO2 is crucial for the production of dry matter and grain. Cropping systems that focus on solely grown crops waste a significant amount of key inputs, such as solar radiation and rainfall, on a seasonal basis. When compared to maize–bean intercropping, both sole crops (maize and beans) experienced a decrease in seasonal ET (ΔET) by 8% and 12% in CON and 5% and 16% under IRWH for maize and beans, respectively. The ET in IRWH increased by 42%, 31%, and 37% compared to CON for sole maize, sole beans, and maize–bean intercrop, respectively (Table 3). This suggests that water losses through crop evapotranspiration were proportional to seasonal runoff and soil evaporation during the growing season. There are three main reasons for the unproductive water losses among treatments: ex-field runoff from the CON in both sole and intercropping, with higher losses during occurrences of few rainstorms; higher water loss through ET in IRWH than CON due to higher infiltration in the IRWH compared to CON tillage; and PAR canopy interception being higher in intercropping compared to sole cropping in both tillage systems (CON and IRWH). In general, WP results were similar to those found in a previous study on maize under IRWH. For example, Passioura [37] and Gregory [57] found that semi-arid ecotopes have a range between 8 and 15 kg ha^−1^ mm^−1^, which is comparable to the IRWH results from this study but higher compared to the CON tillage. This result has important implications for the management practices of IRWH as it confirms the need to optimize water use in terms of yield per unit of water for transpiration to achieve higher WP in water-scarce semi-arid conditions. 

In addition to intercepted PAR, RUE is another important factor for dry matter accumulation. The results in Table 5 show the variation in TPAR and RUE maize–bean sole and intercropping under IRWH and CON tillages. The increase in RUE was consistent with the observed higher rate of increase in above-ground dry matter (Table 4). The maize RUE increased at each subsequent growth stage in both sole and intercropping but was different for different treatments. The RUE for the sole bean was higher than for intercropped beans under IRWH tillage, which was inconsistent with a study conducted by Tsubo et al. [24], where intercropped beans had higher RUE compared with sole beans in conventional tillage with irrigation. Moreover, in Morago, the intercropping under IRWH showed lower TPAR compared to CON. The results showed that in Morago, the IRWH had less TPAR by 25% and 15% compared to CON for intercropped beans and maize, respectively. Similarly, in Paradys, the IRWH showed TPAR values reduced by 15% and 28% compared to CON. This could be related to reduced PAR interception associated with the spatial arrangement of the canopy in the CON and IRWH tillage systems and a reduced canopy size due to water deficit and shading. Studies by Sinclair et al. [58] suggest that intercropped could have higher water stress than sole crops because of the increased evaporative demand of the canopies. However, in semi-arid areas, when there is sufficient water, different growth stages of crops have varying responses to water productivity. For example, Bello et al. [59] discovered that by reducing water usage during times of water deficit, crops can cope with water stress and increase their water WUE.

Water can be stored in the soil in different ways in CON and IRWH, thus differing the use of available resources and crop water demand. The capture of radiation is, in contrast, dependent on canopy structure and planting row arrangements. For example, Hunt et al. [60] have proposed that the resource capture model (dry matter production as a function of the efficiency of resource capture and the efficiency of resource utilization) is useful for all kinds of resources. Monteith [61], however, pointed out that the model is more adequate for radiation and has some restrictions for storable resources. However, irrespective of the adequacy of the model and the contrasting responses for water and radiation, it is possible to develop alternative strategies for the efficient use of resources for smallholder farmers in semi-arid areas. Studies have shown a positive correlation between water use efficiency (WUE) and radiation use efficiency (RUE) under various management practices and environmental conditions. For instance, Singh and Sri Rama [62] found that chickpeas’ RUE positively correlated with soil water content under water-stressed conditions. Similarly, Tsubo et al. [24] reported an increase in maize–bean intercropping systems’ RUE with an increase in WUE. Caviglia et al. [63] also found a close association between WUE and RUE in double-cropped wheat and soybean. 

This study suggests that using TPAR and RUE estimations can improve seasonal water productivity. Other cultural practices in IRWH techniques, such as mulching and growing cover crops, can also help optimize resource use efficiency. A strong relationship between RUE and WP indicates better radiation capture and reduced unproductive water losses. Studies have shown close associations between RUE and WUE for various crops. Increasing radiation capture is an important way to improve water productivity. The positive relationship between RUE and WP indicates a better capture of radiation (Figure 6) in relation to the role of unproductive water losses, mainly via runoff and soil evaporation. There are studies on RUE (for example, Loomis and Connor [64] because RUE is the most widely used indicator of a crop’s ability to use PAR to produce biomass. Close associations between RUE and WUE were reported for sunflower and spring wheat [27], wheat and soybean [65], and maize–bean intercropping [24,65]. Analysis of the relationship between RUE and WP provides insight into how intercropping under improved tillages can increase the seasonal productivity of water or radiation. Increasing radiation capture is an important way to improve water productivity.

## 5. Conclusions

Improving rainfed crop production per unit area for smallholder farmers during the short rainy season is crucial for achieving food and nutrition security. In semi-arid regions, where dry conditions (El Niño seasons) and long dry spells are common due to climate change, farmers are exploring alternative techniques to increase water productivity and use resources more efficiently. Using water harvesting techniques on marginal land can help reduce the pressure to produce grain in less productive and environmentally fragile agroecosystems. The Thaba Nchu rural community faces low productivity and needs improved techniques. However, there is poor adoption of these techniques and disorganized intercropping due to water scarcity in arid and semi-arid areas. This highlights the need for alternative techniques that increase smallholders’ productivity through the ability to capture and use resources more efficiently, particularly water and radiation; although the soil nutrients are not included, the soils can benefit through legume N fixation. The relationships between WP and RUE indicate that the links between the efficiencies in the use of radiation and rainwater remain when upgrading from CON to in situ rainwater IRWH tillage and from solely grown crops to intercropping in a semi-arid environment. To improve WP under IRWH, several options have been proposed, including increasing the HI, the proportion of transpired water, and reducing VPD. However, an effective approach to further enhance water use on a seasonal basis will be to focus on improving the capture of radiation by crops in relation to WP. 

## Figures and Tables

**Figure 1 plants-12-02919-f001:**
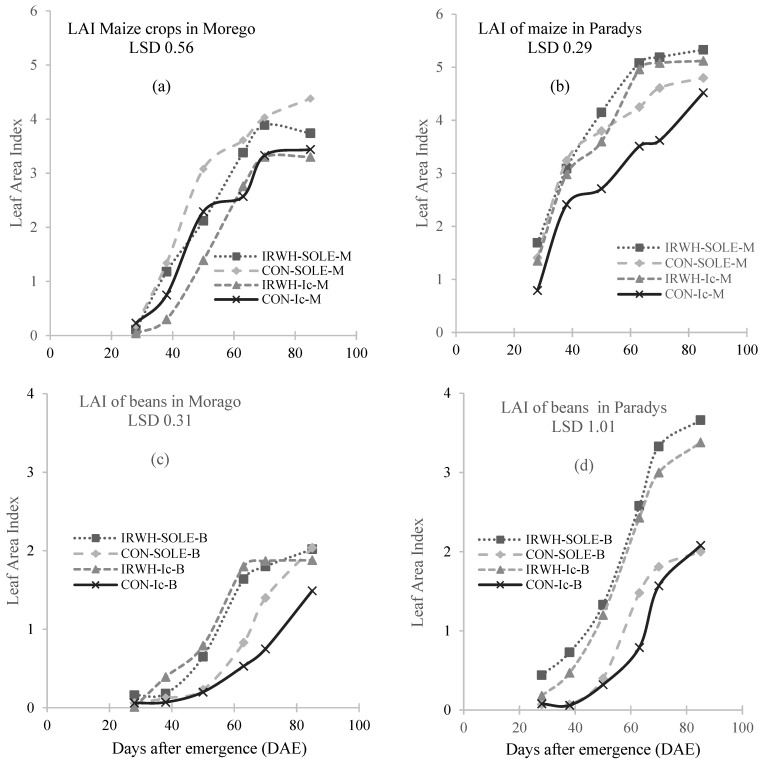
The leaf area index (LAI) for maize (M) and beans (B) growing in three cropping systems (sole-M, sole-B, and intercropping Ic) under two management practices (IRWH and CON) for both Morago (**a**,**c**) and Paradys (**b**,**d**) villages. NB: IRWH-SOLE-M and IRWH-SOLE-B = Sole maize and beans under IRWH tillage. CON-SOLE-M and CON-SOLE-B = Sole maize and beans under CON tillage. IRWH-Ic-M and IRWH-Ic-B = Intercropped maize and beans under IRWH tillage. CON-Ic-M and CON-Ic-B = Intercropped maize and beans under CON tillage.

**Figure 2 plants-12-02919-f002:**
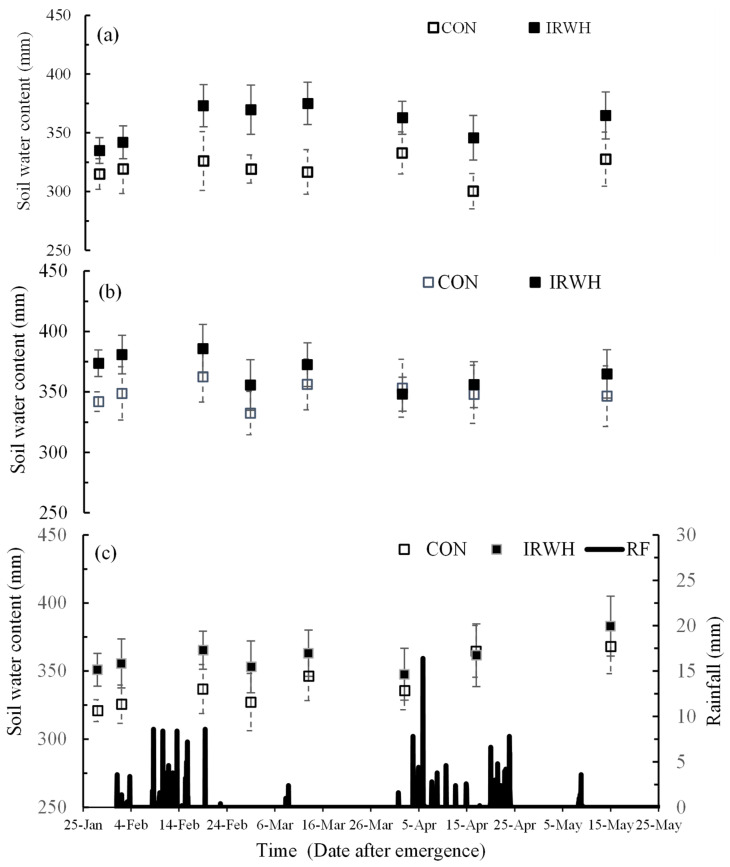
Measured changes in soil water contents of the root zone (0–1500) in the basin area of the IRWH and between plant rows in the CON tillage through the 2018/19 cropping season and daily rainfall (RF) for (**a**) soil maize, (**b**) sole beans, and (**c**) maize–beans intercropping. The error bars represent the standard deviation values.

**Figure 3 plants-12-02919-f003:**
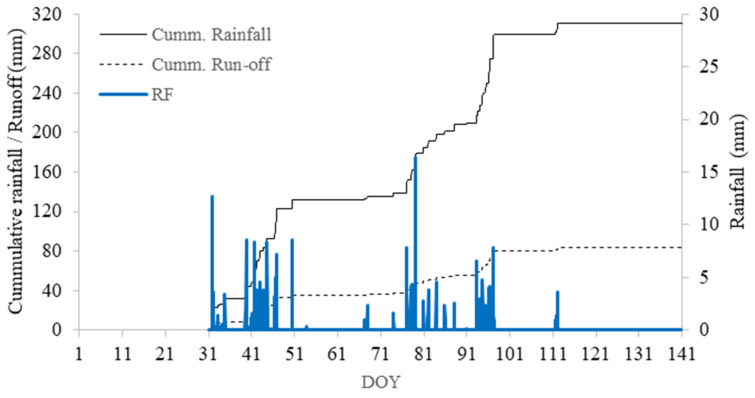
Estimation of in-field runoff (dashed line) from the rainfall (blue bars and full line for cumulative rainfall) during the growing season 2018/19 using an empirical model developed by Anderson (36).

**Figure 4 plants-12-02919-f004:**
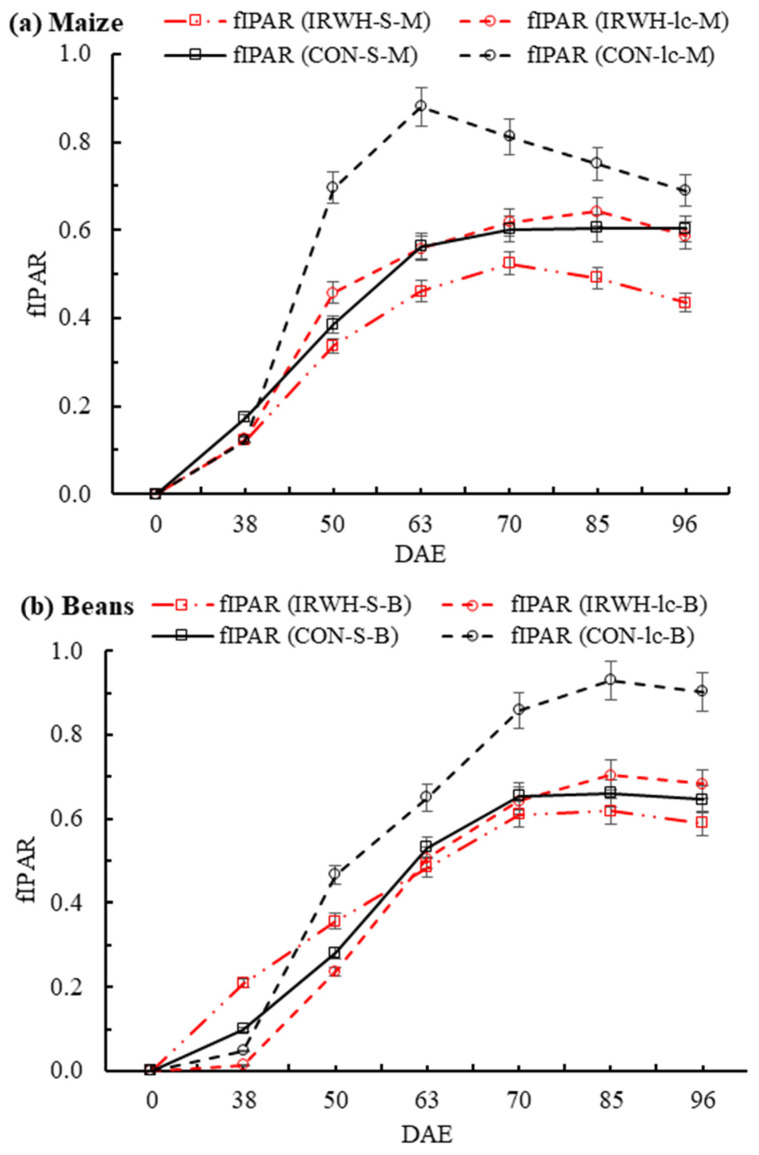
Fraction of intercepted photosynthetic active radiation (fIPAR) of two cropping systems (sole and intercropping) under two tillage systems (CON and IRWH) from the Morago demonstration plots measurement: (**a**) maize and (**b**) beans.

**Figure 5 plants-12-02919-f005:**
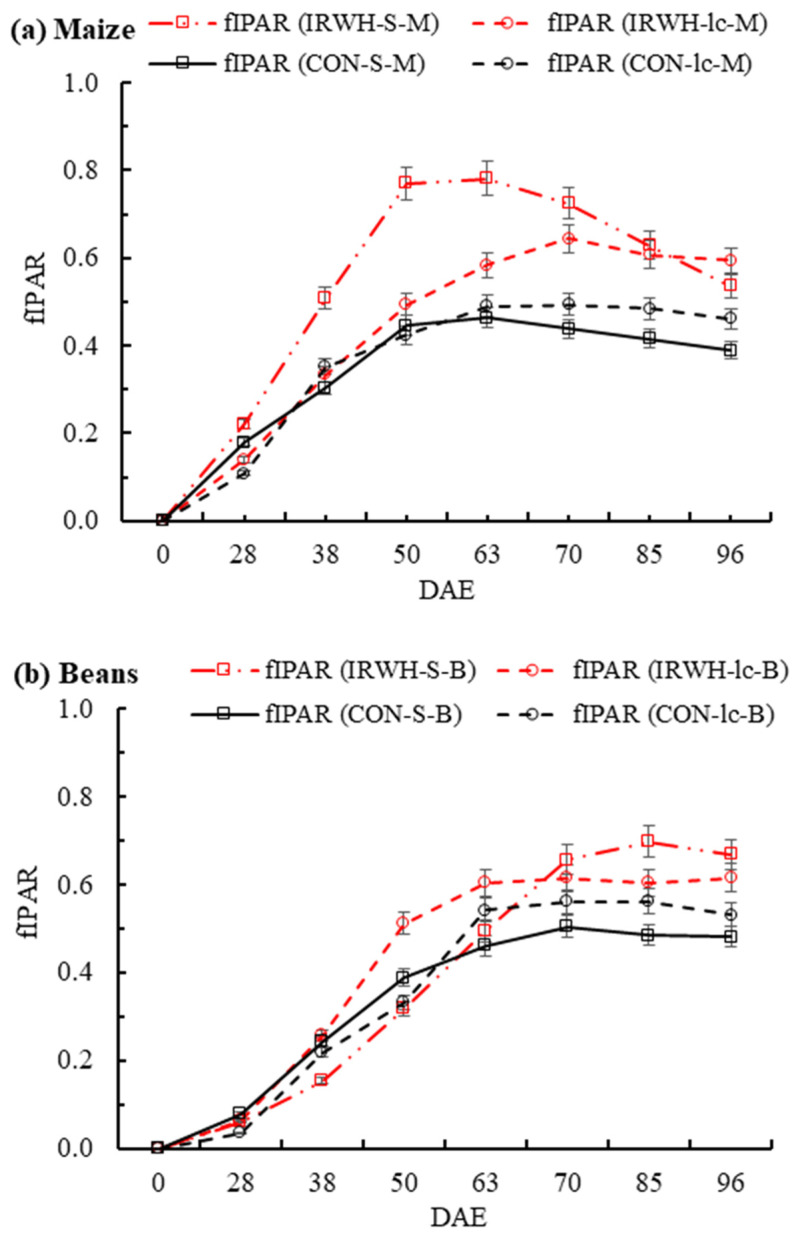
Fraction of intercepted photosynthetic active radiation (fIPAR) of two cropping systems (sole and intercropping) under two tillage systems (CON and IRWH) from the Paradys demonstration plots measurement: (**a**) maize and (**b**) beans.

**Figure 6 plants-12-02919-f006:**
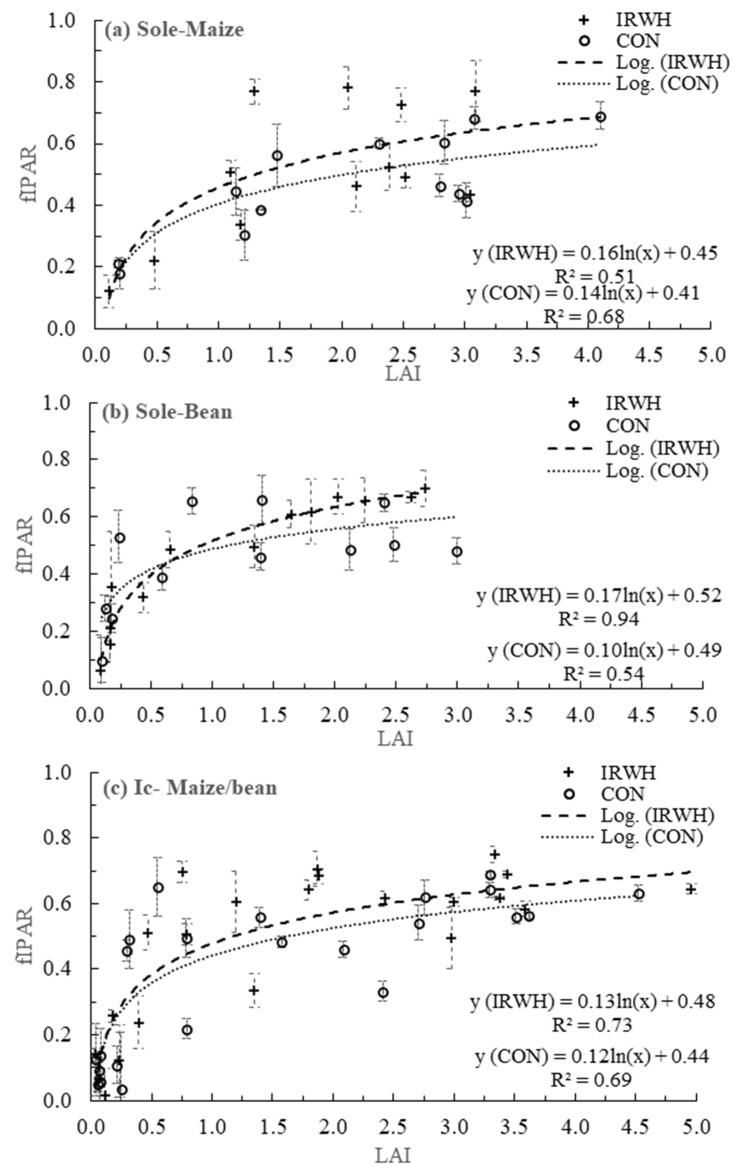
The logarithmic relationship between LAI and fraction intercepted photosynthetic active radiation (fIPAR) for different cropping systems; (**a**) sole maize, (**b**) sole beans, and (**c**) intercropping under CON and IRWH tillage systems.

**Figure 7 plants-12-02919-f007:**
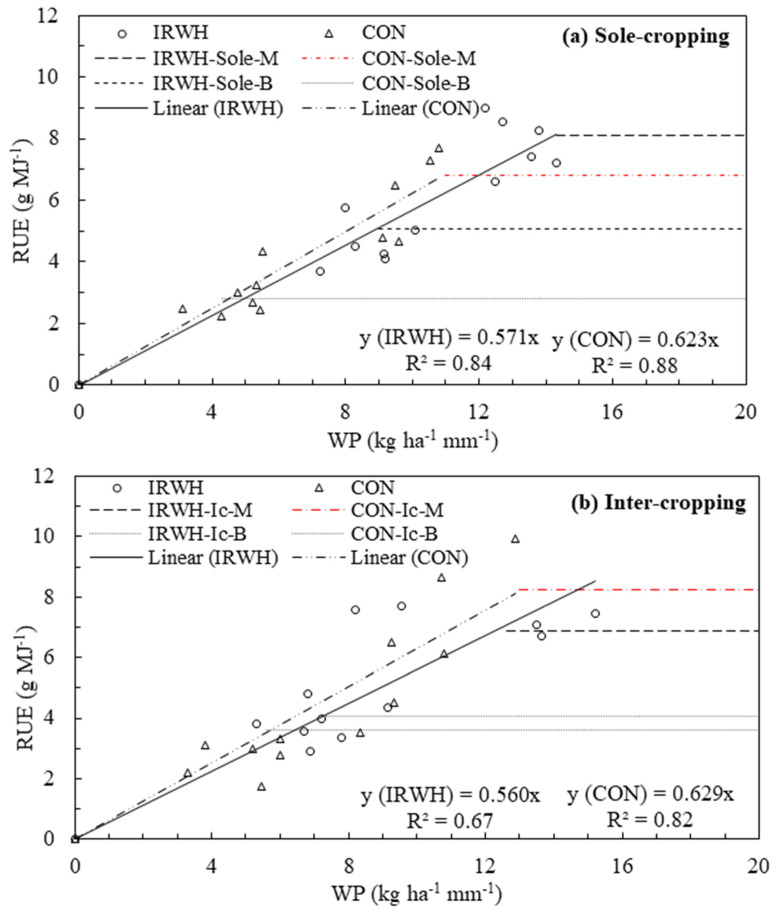
Relationship between water productivity (WP) and radiation use efficiency (RUE) among cropping systems (sole-M, sole-B, and intercropping-Ic) under two different tillage systems (CON and IRWH); WP and RUE were derived as the slope of the regression of AGDM on cumulative water use and radiation use, respectively during the growing season. (**a**) Sole cropping and (**b**) Intercropping.

**Table 1 plants-12-02919-t001:** Total above-ground dry matter (AGDM) accumulation for maize (M) and beans (B) growing in three cropping systems (sole-M, sole-B, and intercropping-Ic) under two management practices (CON and IRWH tillage) for both Morago and Paradys villages.

DAE	Total AGDM, (g m^−2^), (Morago Village)					Total AGDM, (g m^−2^), (Paradys Village)				
	Sole Maize	Maize-Ic	Sole Beans	Beans-Ic	Total M + B	Sole Maize	Maize-Ic	Sole Beans	Beans-Ic	Total M + B
**IRWH**										
28	1.2	0.5	3.8	1.8	2.3	1.8	4.5	2.9	2.2	6.7
38	1.6	0.8	7.5	2.4	3.2	14.1	9.5	12.5	9.1	18.6
50	31.8	4.2	23.4	19.0	23.3	42.1	69.5	48.3	29.9	99.4
63	34.6	21.1	89.2	70.6	91.7	128.4	90.6	109.9	106.4	197.0
70	196.8	88.1	544.6	311.1	399.2	216.2	107.6	534.0	241.0	348.6
85	243.1	220.2	549.2	476.0	696.1	272.8	126.2	642.8	261.2	367.4
96	231.1	205.3	530.6	440.7	646.0	312.1	146.9	738.6	665.4	812.3
**CON**										
28	1.0	2.2	2.1	0.8	3.0	1.4	3.2	1.5	2.8	6.o
38	1.5	3.1	7.2	1.0	4.1	1.6	2.4	1.6	4.5	6.9
50	2.4	20.3	12.8	1.9	4.2	10.4	24.2	10.4	16.3	40.5
63	9.9	23.0	47.4	10.4	33.4	28.8	80.5	28.8	75.8	156.3
70	27.4	76.5	78.7	91.1	167.6	241.1	163.2	241.1	149.5	312.3
85	229.1	155.1	166.5	326.6	481.7	265.7	252.8	265.7	248.9	501.7
96	252.4	240.2	428.3	449.6	689.7	287.9	270.2	288.0	292.1	562.3

**Table 2 plants-12-02919-t002:** Seasonal evapotranspiration (ET) as calculated from the change of soil water content (ΔSW) and in-field runoff (Roff) and for different cropping systems (Sole- maize, Sole-beans and intercropping under (**a**) CON and (**b**) IRWH tillage systems for the growing season of 2018/19.

**Treatments**	**DAE**	**1–28**	**29–38**	**39–50**	**51–63**	**64–70**	**71–85**	**85–121**	**Total**
	P (mm)	14.2	46.2	71.2	19.8	46.6	101.4	11.2	310.6
**(a) CON Tillage**	Run-off (R_off_)	3.8	12.4	19.1	5.3	12.5	27.2	3	−83.2
Sole-Maize	ΔSW	4.4	6.7	−6.8	−2.5	16.2	−32.6	27.3	12.7
	ET	14.8	40.5	45.3	12.0	50.3	41.6	35.5	240
Sole-Beans	ΔSW	6.8	13.8	−30.2	23.8	−3.1	−5.1	−1.6	4.4
	ET	17.2	47.6	21.9	38.3	31.0	69.1	6.6	231.7
Ic-Maize/beans	ΔSW	4.7	11.1	−9.6	19.1	−10.6	16.9	3.5	35.1
	ET	15.1	44.9	42.5	33.6	23.5	91.1	11.7	262.4
**(b) IRWH Tillage**	Run-off (Roff)	0	0	0	0	0	0	0	0
Sole-Maize	ΔSW	6.9	31.1	−3.4	5.3	−12.2	−17.1	19.0	29.8
	ET	21.1	77.3	67.8	25.1	34.4	84.3	30.2	340.4
Sole-Beans	ΔSW	7.1	5.0	−30.1	16.8	−24.4	8.0	8.9	-8.8
	ET	21.3	51.2	41.1	36.6	22.2	109.4	20.1	301.8
Ic-Maize/beans	ΔSW	4.6	9.7	−12.3	10.0	−15.5	31.0	21.4	49.0
	ET	18.8	55.9	58.9	29.8	31.1	132.4	32.6	359.6

NB: The − and + sign for R_off_ and R_on_ represents the ex-field runoff losses from CON and the amount of rainwater harvested in the basin area of IRWH tillage. P, ΔSW and ET represent precipitation, change in soil water content, and evapotranspiration, respectively.

**Table 3 plants-12-02919-t003:** Different water use indicators efficiency (for maize (M) and beans (B) growing in three cropping systems sole-M, sole-B, and intercropping (Ic) under two tillage systems (CON and IRWH). (**a**) WP (kg ha^−1^ mm^−1^) and (**b**) WUE (kg ha^−1^ mm^−1^). AGDM and Yg indicate above-ground dry matter and grain yield.

Treatment	Maize (kg ha^−1^)		Treatment	Beans (kg ha^−1^)	
	AGDM	Yg		AGDM	Yg
**(a) WP**					
IRWH-Sole-M	12.70b	3.73a	IRWH-Sole-B	10.10a	2.83a
IRWH-Ic-M	15.12a	3.53b	IRWH-Ic-B	7.86b	2.51a
CON-Sole-M	9.58c	2.67b	CON-Sole-B	5.43c	2.21a
CON-Ic-M	8.34c	2.63b	CON-Ic-B	5.44c	1.99a
LSD	2.21	1.01	LSD	1.95	0.91
**(b) WUE**					
IRWH-Sole-M	11.59a	3.41a	IRWH-Sole-B	10.40a	2.91a
IRWH-Ic-M	13.06a	3.05a	IRWH-Ic-B	6.79b	2.17a
CON-Sole-M	12.40a	3.46a	CON-Sole-B	7.28b	2.97a
CON-Ic-M	9.87b	3.12a	CON-Ic-B	6.44b	2.36a
LSD	1.68	1.22	LSD	2.13	1.08

Means followed by the same letter are not significantly different (*p* ≤ 0.05).

**Table 4 plants-12-02919-t004:** Total intercepted photosynthesis active radiation TPAR and radiation use efficiency (RUE) and the percentage of improved RUE in intercropping relative to sole cropping under two tillage systems: (**a**) Morago and (**b**) Paradys sites.

DAE	TPAR (MJ)				RUE (AGDM/TPAR, (g MJ^−1^)					RUE Improve (%)
	Maize-Sole	Maize -Ic	Beans-Sole	Beans-Ic	Maize-Sole	Maize -Ic	Beans-Sole	Beans-Ic	Total M + B	
**(a) Morago**										
**IRWH**										
28	-	-	-	-	-	-	-	-	-	-
38	41.1	42.2	70.7	5.1	0.04	0.02	0.11	0.47	0.49	−89.5
50	146.3	198.9	155.0	102.8	0.22	0.02	0.15	0.19	0.21	−26.0
63	258.9	313.1	272.2	284.5	0.13	0.07	0.33	0.25	0.32	8.4
70	355.5	422.0	416.2	438.4	0.55	0.21	1.31	0.71	0.92	9.4
85	373.1	486.5	469.1	534.2	0.65	0.45	1.17	0.89	1.34	17.9
96	385.9	521.3	522.5	606.4	0.60	0.39	1.02	0.73	1.12	20.1
**CON**										
28	-	-	-	-	-	-	-	-	-	-
38	58.4	41.0	33.4	16.3	0.03	0.07	0.21	0.06	0.14	−27.9
50	167.5	302.5	122.2	202.6	0.01	0.01	0.10	0.01	0.02	81.8
63	316.2	494.1	298.3	365.0	0.03	0.05	0.16	0.03	0.07	46.4
70	410.7	554.7	446.7	585.6	0.07	0.14	0.18	0.16	0.29	31.1
85	456.4	568.7	500.8	705.5	0.50	0.27	0.33	0.46	0.74	34.4
96	534.0	611.8	577.1	800.4	0.47	0.39	0.74	0.56	0.95	28.2
**(b) Paradys**										
**IRWH**										
28	74.8	47.2	20.6	21.5	0.02	0.09	0.14	0.10	0.20	−9.5
38	220.5	145.2	66.9	111.9	0.06	0.07	0.19	0.08	0.15	10.8
50	432.1	277.5	178.4	287.3	0.10	0.25	0.27	0.10	0.35	9.1
63	532.5	398.5	337.8	413.1	0.24	0.25	0.33	0.26	0.51	1.9
70	549.8	488.6	498.4	466.2	0.39	0.14	1.07	0.52	0.66	−8.4
85	557.3	539.3	619.7	536.6	0.49	0.20	1.04	0.49	0.68	−10.5
96	525.1	582.0	655.9	604.0	0.59	0.25	1.13	1.10	1.35	−2.5
**CON**										
28	60.4	36.5	26.5	12.1	0.02	0.09	0.05	0.27	0.36	−50.7
38	131.8	153.0	106.0	95.1	0.01	0.02	0.02	0.03	0.04	2.9
50	250.4	237.9	218.3	186.6	0.04	0.10	0.05	0.02	0.12	0.2
63	316.5	334.4	314.6	369.5	0.09	0.24	0.09	0.22	0.46	11.4
70	332.0	374.7	382.3	425.6	0.73	0.44	0.63	0.38	0.82	8.7
85	368.6	429.7	430.6	498.9	0.72	0.59	0.62	0.51	1.10	12.2
96	381.9	452.0	471.9	521.4	0.75	0.60	0.61	0.52	1.12	9.1

**Table 5 plants-12-02919-t005:** Summary of the results from Figure 7a,b showing maximum RUE, R^2^, and the regression slope for different treatments with sole- and intercropping systems.

Treatments	Sole-Cropping			Treatments	Intercropping		
	Max. RUE (g MJ^−1^)	R^2^	Slope		Max. RUE (g MJ^−1^)	R^2^	Slope
IRWH-Sole-M	8.12	0.84	0.571	IRWH-Ic-M	6.89	0.67	0.560
IRWH-Sole-B	5.09			IRWH-IC-B	4.05		
CON-Sole-M	7.16	0.88	0.623	CON-Ic-M	8.46	0.82	0.629
CON-Sole-B	2.81			CON-Ic-B	3.60		

## Data Availability

Data is contained within the article.

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
