# Peer review of "In-Field Rainwater Harvesting Tillage in Semi-Arid Ecosystems: II Maize–Bean Intercrop Water and Radiation Use Efficiency"

_plants, 2023, doi:10.3390/plants12162919_

Round 1
Reviewer 1 Report
There is no description or definition of the two main effects, CON and IRWH. You cannot assume that all readers are familiar with the types of production used here. The paper is far to long for the amount of information presented. Most of the figures and tables are superfluous.
Author Response
Dear Reviewer
Comments:
There is no description or definition of the two main effects, CON and IRWH. You cannot assume that all readers are familiar with the types of production used here. The paper is far to long for the amount of information presented. Most of the figures and tables are superfluous.
Response:
The authors are working to improve the MS.
- The two tillage systems (CON and IRWH) are described in both the introduction and methodology sections. And clearly abbreviated on both the abstract and the main text body.
- The paper is quite long, but the main reason for using many results from the farmer's field measurements that demonstrate the difference in two villages with different soils under different tillage systems as well as cropping systems. The authors agree to present results from growth, Yield, BM and PUE parameres, etc. This will help to explain the practical measurement from on-farm demonstration plots while farmers participated and engaged in the project/study directly. This is clearly explained in the study site description and farmers' selection on the study.
- All editorial corrections are done and improved the abstract section and revise the whole conclusion section.
- Please see attached some major changes made to the MS. And some editorial language corrections are made in the MS.
Thank you!

Reviewer 2 Report
Line # 14- Don’t abbreviate for intercropping
Line 14: use word in-situ moisture conservation in place of in-field rainwater harvesting
Line 16 consistent use terminologies line 17 and 18 use either plot or treatment for example treatment vs plot, solely v/s sole; beans sole vs sole beans
Line 24 delete solely grown maize replace with sole maize
Line 25 and 26 revise sentence
Line 28 are replaced with were
Line 32 radiation use efficiency replaced by RUE since it is already abbreviated in line 20.
Line 32,33 and 34 delete entire sentence
Arrange key words according alphabet
Throughout the text replace in-field word with In-situ
Line 60 cite recent reference
Line 72 delete radiation use efficiency replaced by RUE since it is already abbreviated in line 20.
Sentence is incomplete Iine 72 and 73
Line 78 CON is already abbreviated in line #17
Revise last paragraph of introduction for meaningful sentences
Line 178 delete the word ‘in the wavelength’
Delete sentence 232- 233
Delete sentence in 312- 314.
Line # 331 Delete word secondly
Quote cite for maize and beans growth phases (Line #353- 358).
Mention line quantum sensor manufacturer name and product number; is it one meter length
Mention the indication of TLM and hm
Statistical analysis part is missed in the text.
Table 3 footnote expand AGDM, Yg
Give proper reasons for Fig.5 intercepted PAR in sole maize was higher than intercropping?
In general intercropping LAI will be more than sole crop.
Table 4 RUE % improve, is it over sole maize or sole beans?
RUE of maize is too low?
Citation missing in reference section Monteith in Line #213
Ndakidemi, 2006 should be as Ndakidemi et al., 2006
Line 73 sadras et al. 2006 should be replaced with Sadras (2006)
Sinclair 1983, missed in reference section
Line 688 Tsubo et al., 2004 should be written as Tsubo and walker 2004
Line 337 Walker and Tsubo 2003a missed in reference
Discussion section should be improved with proper data and % improvement in intercropping over sole crops.
Line # 659 is contradicting to your results sole maize and intercropped maize.
Conclusion is too elaborative try to be in precise, avoid quoting figure numbers, references in the conclusion
References
Reference 3, 12, 18, 20, 24, 28, 29, 41, 53 were not found in text
Reference 47 repeated
Reference 49 and 51 authors and year are same; separate them by ‘a’ and ‘b’
Author Response
Reviewer #2 comment Response
Line # 14- Don’t abbreviate for intercropping
- Response: On the abstract: deleted "(Ic)
Line 14: use the word in-situ moisture conservation in place of in-field rainwater harvesting
- Response: The term IRWH = is a common name known in dryland farming for smallholders to harvest water from in-field to minimize avoid runoff. So it is difficult to use the in-situ moisture conservation
Line 16 consistent use terminologies line 17 and 18 use either plot or treatment for example treatment vs plot, solely v/s sole; beans sole vs sole beans;
- Response: deleted and changed to plot and sub-plot
Line 24 delete solely grown maize replace with sole maize:
- Response: changed accordingly except in a few sentences to fit in to make specification.
Line 25 and 26 revise sentence:
- Response: It is rephrased as "The RUE under IRWH tillage was estimated to be 0.65 and 0.39 g DM MJ-1, in sole maize and intercropping respectively.
Line 28 are replaced with were
Line 32 radiation use efficiency replaced by RUE since it is already abbreviated in line 20.
- Response= Corrected by "RUE"
Line 32,33 and 34 delete entire sentence:
- Response = Deleted the sentence and rephrased the following sentence
Arrange key words according alphabet:
- Response = re-arrange the key words accordingly "
Keywords: Evapotranspiration; radiation interception; radiation use efficiency; soil water balance; water productivity
Throughout the text replace in-field word with In-situ:
- Response: In this study, it seems inappropriate to change the terminology as it is known as IRWH for the 2-3 decades in this type of environment in SA. But Explained in methodology the stsrucut=re and explained in the above.
Line 60 cite recent reference:
- Response: well taken comment In general the introduction section is modified by updating the references and adding relevant background information (see the highlight sections)
Line 72 delete radiation use efficiency replaced by RUE since it is already abbreviated in line 20.
- Response: replaced as "RUE"
Sentence is incomplete Iine 72 and 73
- Response: rephrased and added verb
Line 78 CON is already abbreviated in line #17
- Response = corrected as "CON"
Revise last paragraph of introduction for meaningful sentences:
- Response: The whole paragraph is repharsed and changed and the last sentence is changed as: "This study, therefore, hypothesizes that maize-bean intercropping under the Improved tillage (IRWH) system increases resource productivity and efficiency compared to solely grown crops. Additionally, there are positive relationships between WP and RUE in both IRWH and CON tillage systems, with higher water deficit and lesser available radiation use in CON compared to IRWH."
Line 178 delete the word ‘in the wavelength = it is deleted ’
Delete sentence 232- 233 = deleted
Delete sentence in 312- 314. = Deleted
Line # 331 Delete word secondly = deleted and rephrased
Quote cite for maize and beans growth phases (Line #353- 358).
- Response = It is referenced as"
The growth stages of the maize and beans described in this study are almost identical to other previous studies (for example that reported by FAO: Doorenbos and Kassam, 1986; FAO, 2000). For both crops, the growth stages can be divided into four phases:
- For maize - GS-1 = initial vegetative phase, GS-2 = active vegetative phase, GS=3 initial grain-filling phase, and GS-4 = active grain-filling phase.
- For beans – GS-1 = emergence and early vegetative growth, GS-2 branching and rapid vegetative growth, GS-3 = flowering & pod formation and GS-4 = pod fill and maturation
Mention line quantum sensor manufacturer name and product number; is it one meter length
- Response: Addedd - the model and manufacturer "(Kipp & Zonen model PQS 1, LI-COR models LI-190)"
Mention the indication of TLM and hm
- Response: Where TLM is the total maize leaf area, and hM and hB are the height of maize and beans canopy."Response : It has already indicated "
Statistical analysis part is missed in the text.
- Response: Added the following as statisy=tical analysis
Analysis of variance (ANOVA) was done for the comparison of different treatments using SAS 9.1.3 for Windows (SAS Inst Inc., 2006). When the significance of the treatment on the F-statistic is mentioned, it refers to a comparison using the least significant differences (LSD) at the 0.05 probability level. In the study, a relationship between RUE and WP was analyzed as the slope of the linear regression using aggregated data from the two locations for different cropping systems under different tillage to understand the effect of available soil water for productivity and the atmospheric demand in the semi-arid crop production system."
- Response:
Table 3 footnote expand AGDM, Yg =
- Response: Added and corrected all GY and changed to Yg = for grain yield.
Give proper reasons for Fig.5 intercepted PAR in sole maize was higher than intercropping.
In general intercropping LAI will be more than sole crop.
Table 4 RUE % improve, is it over sole maize or sole beans?
Response: Added in the methodology about Change in RUE:
Where WB and WM are dry matter (in kg) for beans and maize, respectively and Io is the incident radiation in (MJ m-2 d-1). Comparative change in RUE was calculated according to Morris and Garrity (1993) to relate productivity across varying intercropping periods and densities (D). The indice was based on relative rather than absolute values. Change in RUE was calculated based on dry matter as shown in equation 13.
see equation 13 in the text
where X is RUE, Pm is the proportion of maize in intercrop, Pb is the proportion of beans in intercrop, subscripts Ic is intercrop, sm is sole maize, and sb is sole beans. Proportions of maize and beans in the intercrop, given by Pm = Dm/(Dm + Db) with Dm and Db being the density in intercropping relative to sole cropping of maize and beans, respectively. For interpretation, when Δ’s greater than zero, are assumed to be higher in the intercrop system relative to the sole crop.
RUE of maize is too low?
Response: Yes I agree but the main reason is the spacing of maize under IRWH is wide as included in the empty runoff section which makes the RUE very low. In addition as dry; and farming the plant population is very low (only 18,000/ha). Thus it is expected lower values.
Citation missing in reference section Monteith in Line #213 = corrected
Ndakidemi, 2006 should be as Ndakidemi et al., 2006 = corrected
Line 73 Sadras et al. 2006 should be replaced with Sadras (2006) = corrected
Sinclair 1983, missed in reference section = corrected
Line 688 Tsubo et al., 2004 should be written as Tsubo and walker 2004 = corrected
Line 337 Walker and Tsubo 2003a missed in reference = corrected
Discussion section should be improved with proper data and % improvement in intercropping over sole crops.
- Response: As suggested the discussion section is re-arranged and some sections are moved from the result section and corrected accordingly. see the new discussion section in the text.
Line # 659 is contradicting to your results sole maize and intercropped maize.
- Response:- Deleted and corrected the contradictory statement but the authors presented the findings for each village with some inconsistent results for IRWH-S- M during 38-70 DAE and for IRWH -S-B at the later stage (70-96 DAE). As the experiment was conducted during dry season there was some variation in selling emergence and growth etc.
Conclusion is too elaborative try to be in precise, avoid quoting figure numbers, references in the conclusion
- Response: the whole conclusion is re-write again and edited as:
Improving rainfed crop production per unit area for smallholder farmers during the short rainy season is crucial for achieving food and nutrition security. In semi-arid regions, where dry conditions (El Niño seasons) and long dry spells are common due to climate change, farmers are exploring alternative techniques to increase water productivity and use resources more efficiently. Using water harvesting techniques on marginal land can help reduce the pressure to produce grain in less productive and environmentally fragile agroecosystems. The Thaba Nchu rural community faces low productivity and needs improved techniques. However, there is poor adoption of these techniques and disorganized intercropping due to water scarcity in arid and semi-arid areas. This highlights the need for alternative techniques that increase smallholders’ productivity through the ability to capture and use resources more efficiently, particularly water and radiation, although the soil nutrients are not included the soils can benefit through legume N fixation ability. The relationships between WP and RUE indicate that the links between the efficiencies in the use of radiation and rainwater remain when upgrading from CON to in-situ rainwater IRWH tillage and from solely grown crops to intercropping in a semi-arid environment. To improve WP under IRWH, several options have been proposed including increasing the HI, the proportion of transpired water, and reducing VPD. However, the effective approach to further enhance water use on a seasonal basis would be to focus on improving the capture of radiation by crops in relation to WP.
References
Reference 3, 12, 18, 20, 24, 28, 29, 41, 53 were not found in text
-Response: Reference 47 repeated: Remove the references as confused with PART-I. Remove ane edited the references and added new updated references
Reference 49 and 51 authors and year are same; separate them by ‘a’ and ‘b’

Reviewer 3 Report
The presented manuscript is the second part of the two manuscripts focusing on the effect of different management practices on production, water and radiation use efficiencies in three cropping systems. This manuscript includes data mainly about water and radiation use efficiencies. The presented manuscript contains well designed experiment with many results.
However, I have some recommendations mainly for Results and Discussion section.
Introduction
This part is clear and sufficient.
Materials and Methods
You should thoroughly check all parameters and unify them throughout the whole text, tables, and figures, e.g. parameter Yg is not defined (probably Yield used in the other manuscript?); lines 398 and 434: GY (grain yield? the same as Yg?); AGDM vs TDM (the same?); TPAR vs TIPAR vs IPAR.
At the end of the Methods, add the paragraph about statistical analyses used.
Results and Discussion
Some parts of the section Results belong to Discussion:
P7-8 L274-286; P16 L509-510, 515-517; P19 L575-584; P21 L618-626.
Figures
Figs. 1: there is no clear what means “Ic” and other abbreviations in legends. You should describe it in the heading.
For example, Fig 1:
“The leaf area index for maize (M) and beans (B) growing in three cropping systems (sole-M, sole-B, and intercropping – Ic) under two management practices (in-field rainwater harvesting – IRWH and conventional – CON) during the growing season for both Morago (a, c) and Paradys (b, d).”
It is necessary to correct this for the other table and figure headings.
You should also better explain statistical analyses (effect of treatment? F value and significance) used in tables and figures.
Fig. 2: I recommend to merge graphs (x axes are the same) and rainfall could be only in the bottom one.
Fig. 7: You have one red vertical line for maximum value of RUE. If you use two red (full line and dashed) a two black (full line and dashed) it will be clearer.
The unit at y-axis is probably only “g MJ-1”.
Author Response
Reviewer # 3 Response
The presented manuscript is the second part of the two manuscripts focusing on the effect of different management practices on production, water and radiation use efficiencies in three cropping systems. This manuscript includes data mainly about water and radiation use efficiencies. The presented manuscript contains well-designed experiment with many results.
However, I have some recommendations mainly for Results and Discussion section.
Introduction
This part is clear and sufficient.
= Added some updated references according to other reviewers
Materials and Methods
You should thoroughly check all parameters and unify them throughout the whole text, tables, and figures, e.g. parameter Yg is not defined (probably Yield used in the other manuscript?); lines 398 and 434: GY (grain yield? the same as Yg?); AGDM vs TDM (the same?); TPAR vs TIPAR vs IPAR.
- Response = GY = changed to Yg (in all cases
- AGDM use for the final total dry matter while TDM was used DM measured during the growth stages. However, all changes made to "AGDM"
- corrected as TPAR, and changed all TIPAR to TPAR
At the end of the Methods, add the paragraph about the statistical analyses used.
- Response= Added as:
2.5 Statistical analysis
Analysis of variance (ANOVA) was done for the comparison of different treatments using SAS 9.1.3 for Windows (SAS Inst Inc., 2006). When the significance of the treatment on the F-statistic is mentioned, it refers to a comparison using the least significant differences (LSD) at the 0.05 probability level. In the study, a relationship between RUE and WP was analyzed as the slope of the linear regression using aggregated data from the two locations for different cropping systems under different tillage to understand the effect of available soil water for productivity and the atmospheric demand in the semi-arid crop production system.
Results and Discussion
Some parts of the section Results belong to Discussion:
P7-8 L274-286; P16 L509-510, 515-517; P19 L575-584; P21 L618-626.
- Response - All moved to discussion section and other comments added. In general, the discussion section is re-arranged and re-write some paragrapghs
Figures
Figs. 1: there is no clear what means “Ic” and other abbreviations in legends. You should describe it in the heading.
It is explained in the cation description as commented and added legend
For example, Fig 1:= Yes = Thank you, well taken comment
“The leaf area index for maize (M) and beans (B) growing in three cropping systems (sole-M, sole-B, and intercropping – Ic) under two management practices (in-field rainwater harvesting – IRWH and conventional – CON) during the growing season for both Morago (a, c) and Paradys (b, d).”
It is necessary to correct this for the other table and figure headings.
-Response: all captions descriptions corrected accordingly
You should also better explain statistical analyses (effect of treatment? F value and significance) used in tables and figures.
- Response: Statistical analysis section/paragraph added
Fig. 2: I recommend merging graphs (x axes are the same) and rainfall could be only in the bottom one.
- Response: corrected as mentioned in the comment = use only one x-axes
Fig. 7: You have one red vertical line for maximum value of RUE. If you use two red (full line and dashed) a two black (full line and dashed) it will be clearer.
The unit at y-axis is probably only “g MJ-1”. = corrected

Reviewer 4 Report
Dear authors, your manuscript addresses an interesting topic. However, major changes are needed in order to improve its structure and presentation.
Kind regards

Author Response
Reviewer # 4 comments from the pdf file
Comment L15-16- During the typical drought season ..... deleted
Comment L17 - the main treatment changed to the main plot
Comment Line 35 - the keywords are re-arranged and edited
Comment L38 - -45; 55-56 and other parts - The introduction section - updated with recent references see some changes made:
Introduction- section - corrections
The introduction section - updated with recent references added
Improving both water use efficiency (WUE) and radiation use efficiency (RUE) in mixing cropping systems is crucial for increasing crop yields in dryland agriculture (Maitra, 2020; Pierre, 2022). The crops in an intercropping system use resources differently, complementing each other and collectively producing higher yields than when grown individually in the same area (Maitra and Gitari, 2020; Duvvada and Maitra, 2020).
Cereal-legume intercropping has several benefits, including increased yield (Yin et al., 2018), improved soil properties (Nasar et al., 2020, and increased nitrogen-fixing bacteria (Solanki et al. 2019; Yu et al., 2019). Organized intercropping systems can also make better use of resources such as light, heat, water, and nutrients, resulting in higher yields and more stable crop groups (Fan et al., 2006; Mei et al., 2012). However, to avoid negative impacts on crop growth, factors such as legume species selection, seeding rate, sowing date and row spacing must be considered to limit competition between legumes and primary crops (Lawson et al., 2007).
Deleted section:
, in many African countries and contribute to food and nutrition security in the livelihood of smallholders. Canopy structures and root systems of cereal crops are generally different from those of legume crops.
- The last paragraph of the introduction section changed as:
This study, therefore, hypothesizes that maize-bean intercropping under the Improved tillage (IRWH) system increases resource productivity and efficiency compared to solely grown crops. Additionally, there are positive relationships between WP and RUE in both IRWH and CON tillage systems, with higher water deficit and lesser available radiation use in CON compared to IRWH.
Comment L56 - added new references at the beginning of the section.
Materials and method
- Comment L92 : corrected as 'is situated at a latitude of 29o 12’33.6”S, Longitude 26o 50’20.3”E, and Altitude of 1516 m.....
- Comment Line 102 - Soil physical-chemical property.: Added a sentence however the details of the soils information presented in the MS - Part-I with all physical, chemical and morphological data and descriptions;- Added:The clay loam soils of the demonstration plots belong to Sapane ecotope. The basic soil morphological properties are deep dark brown and brown-grey-black, for Paradys and Morago with A horizon of clay loam having a particle size of clay 34.0% and 29.4%, respectively.
- Comment L304-306 = the x-axis corrected and use only one at the bottom
Discussion section
- Comment: L509 510: moved to the discussion section and added information about interception and photosynthesis
- Comment L628 ---634 The Discussion section. Some parts of the result sections moved to Discussion some parts were re-write and added information and references (see the revised MS).
Conclusion section
- Comment L694--718: The conclusion section - modified and corrected and re-write the whole part and avoided the referencing: see below the new conclusion section
"Improving rainfed crop production per unit area for smallholder farmers during the short rainy season is crucial for achieving food and nutrition security. In semi-arid regions, where dry conditions (El Niño seasons) and long dry spells are common due to climate change, farmers are exploring alternative techniques to increase water productivity and use resources more efficiently. Using water harvesting techniques on marginal land can help reduce the pressure to produce grain in less productive and environmentally fragile agroecosystems. The Thaba Nchu rural community faces low productivity and needs improved techniques. However, there is poor adoption of these techniques and disorganized intercropping due to water scarcity in arid and semi-arid areas. This highlights the need for alternative techniques that increase smallholders’ productivity through the ability to capture and use resources more efficiently, particularly water and radiation, although the soil nutrients are not included the soils can benefit through legume N fixation ability. The relationships between WP and RUE indicate that the links between the efficiencies in the use of radiation and rainwater remain when upgrading from CON to in-situ rainwater IRWH tillage and from solely grown crops to intercropping in a semi-arid environment. To improve WP under IRWH, several options have been proposed including increasing the HI, the proportion of transpired water, and reducing VPD. However, the effective approach to further enhance water use on a seasonal basis would be to focus on improving the capture of radiation by crops in relation to WP."
References:
Added and updated with recent citations in the subject matter of WUE/RUE of intercropping.... see the revised MS with yellow highlights....

Round 2
Reviewer 1 Report
Paper is much improved.
Reviewer 4 Report
None